# 1000+ FPS 4D Gaussian Splatting for Dynamic Scene Rendering

Yuheng Yuan[1,*]    Qiuhong Shen[1,*]    Xingyi Yang[2,1]    Xinchao Wang[1,†]

[1]National University of Singapore    [2]The Hong Kong Polytechnic University

{yuhengyuan,qiuhong.shen}@u.nus.edu, xingyi.yang@polyu.edu.hk, xinchao@nus.edu.sg

*Project page:* https://4dgs-1k.github.io/

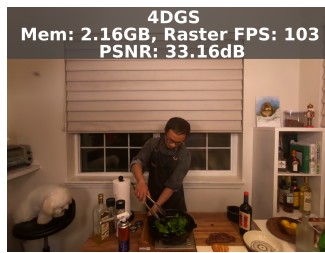 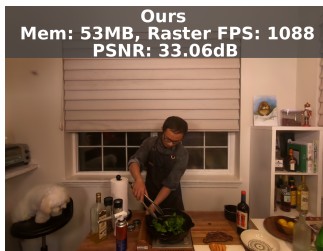 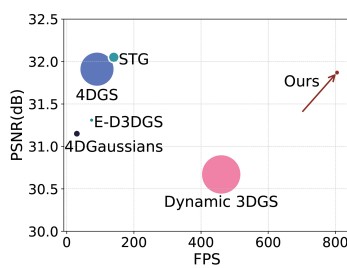

Figure 1: **Compressibility and Rendering Speed.** We introduce **4DGS-1K**, a novel compact representation with high rendering speed. In contrast to 4D Gaussian Splatting (4DGS) [1], we can achieve rasterization at 1000+ FPS while maintaining comparable photorealistic quality with only 2% of the original storage size. The right figure is the result tested on the N3V [2] datasets, where the radius of the dot corresponds to the storage size.

## Abstract

4D Gaussian Splatting (4DGS) has recently gained considerable attention as a method for reconstructing dynamic scenes. Despite achieving superior quality, 4DGS typically requires substantial storage and suffers from slow rendering speed. In this work, we delve into these issues and identify two key sources of temporal redundancy. (Q1) **Short-Lifespan Gaussians**: 4DGS uses a large portion of Gaussians with short temporal span to represent scene dynamics, leading to an excessive number of Gaussians. (Q2) **Inactive Gaussians**: When rendering, only a small subset of Gaussians contributes to each frame. Despite this, all Gaussians are processed during rasterization, resulting in redundant computation overhead. To address these redundancies, we present **4DGS-1K**, which runs at over 1000 FPS on modern GPUs. For Q1, we introduce the Spatial-Temporal Variation Score, a new pruning criterion that effectively removes short-lifespan Gaussians while encouraging 4DGS to capture scene dynamics using Gaussians with longer temporal spans. For Q2, we store a mask for active Gaussians across consecutive frames, significantly reducing redundant computations. Compared to vanilla 4DGS, our method achieves a $41\times$ reduction in storage and $9\times$ faster rasterization on complex dynamic scenes, while maintaining comparable visual quality.

## 1 Introduction

Novel view synthesis for dynamic scenes allows for the creation of realistic representations of 4D environments, which is essential in fields like computer vision, virtual reality, and augmented reality. Traditionally, this area has been led by neural radiance fields (NeRF) [2–6], which model opacity and

---

*Equal contribution.
†Corresponding Author.

39th Conference on Neural Information Processing Systems (NeurIPS 2025).

color over time to depict dynamic scenes. While effective, these NeRF-based methods come with high training and rendering costs, limiting their practicality, especially in real-time applications and on devices with limited resources.

Recently, point-based representations like 4D Gaussian Splatting (4DGS) [1] have emerged as strong alternatives. 4DGS models a dynamic scene using a set of 4D Gaussian primitives, each with a 4-dimensional mean and a $4 \times 4$ covariance matrix. At any given timestamp, a 4D Gaussian is decomposed into a set of conditional 3D Gaussians and a marginal 1D Gaussian, the latter controlling the opacity at that moment. This mechanism allows 4DGS to effectively capture both static and dynamic features of a scene, enabling high-fidelity dynamic scene reconstruction.

However, representing dynamic scenes with 4DGS is both storage-intensive and slow. Specifically, 4DGS often requires millions of Gaussians, leading to significant storage demands (averaging 2GB for each scene on the N3V [2] dataset) and suboptimal rendering speed. In comparison, mainstream deformation field methods [7] require only about 90MB for the same dataset. Therefore, reducing the storage size of 4DGS [1] and improving rendering speed are essential for efficiently representing complex dynamic scenes.

We look into the cause of such an explosive number of Gaussian and place a specific emphasis on two key issues. **(Q1)** A large portion of Gaussians exhibit a short temporal span. In empirical experiments, 4DGS tends to favor "flicking" Gaussians to fit complex dynamic scenes, which just influence a short portion of the temporal domain. This necessitates that 4DGS relies on a large number of Gaussians to reconstruct a high-fidelity scene. As a result, substantial storage is needed to record the attributes of these Gaussians. **(Q2)** Inactive Gaussians lead to redundant computation. During rendering, 4DGS needs to process all Gaussians. However, only a very small portion of Gaussians are active at that moment. Therefore, most of the computation time is spent on inactive Gaussians. This phenomenon greatly hampers the rendering speed. In this paper, we introduce **4DGS-1K**, a framework that significantly reduces the number of Gaussians to minimize storage requirements and speedup rendering while maintaining high-quality reconstruction. To address these issues, 4DGS-1K introduces a two-step pruning approach:

- **Pruning Short-Lifespan Gaussians.** We propose a novel pruning criterion called the *spatial-temporal variation score*, which evaluates the temporal impact of each Gaussian. Gaussians with minimal influence are identified and pruned, resulting in a more compact scene representation with fewer Gaussians with short temporal span.

- **Filtering Inactive Gaussians.** To further reduce redundant computations during rendering, we use a key-frame temporal filter that selects the Gaussians needed for each frame. On top of this, we share the masks for adjacent frames. This is based on our observation that Gaussians active in adjacent frames often overlap significantly.

Besides, the pruning in step 1 enhances the masking process in step 2. By pruning Gaussians, we increase the temporal influence of each Gaussian, which allows us to select sparser key frames and further reduce storage requirements.

We have extensively tested our proposed model on various dynamic scene datasets including real and synthetic scenes. As shown in Figure 1, 4DGS-1K reduces storage costs by $41\times$ on the Neural 3D Video datasets [2] while maintaining equivalent scene representation quality. Crucially, it enables real-time rasterization speeds exceeding 1,000 FPS. These advancements collectively position 4DGS-1K as a practical solution for high-fidelity dynamic scene modeling without compromising efficiency.

In summary, our contributions are three-fold:

- We delve into the temporal redundancy of 4D Gaussian Splatting, and explain the main reason for the storage pressure and suboptimal rendering speed.

- We introduce **4DGS-1K**, a compact and memory-efficient framework to address these issues. It consists of two key components, a spatial-temporal variation score-based pruning strategy and a temporal filter.

- Extensive experiments demonstrate that 4DGS-1K not only achieves a substantial storage reduction of approximately $41\times$ but also accelerates rasterization to 1000+ FPS while maintaining high-quality reconstruction.

## 2 Related Work

### 2.1 Novel view synthesis for static scenes

Recently, neural radiance fields(NeRF) [3] have achieved encouraging results in novel view synthesis. NeRF [3] represents the scene by mapping 3D coordinates and view dependency to color and opacity. Since NeRF [3] requires sampling each ray by querying the MLP for hundreds of points, this significantly limits the training and rendering speed. Subsequent studies [8–15] have attempted to speed up the rendering by introducing specialized designs. However, these designs also constrain the widespread application of these models. In contrast, 3D Gaussian Splatting(3DGS) [16] has gained significant attention, which utilizes anisotropic 3D Gaussians to represent scenes. It achieves high-quality results with intricate details, while maintaining real-time rendering performance.

### 2.2 Novel view synthesis for dynamic scenes

Dynamic NVS poses new challenges due to the temporal variations in the input images. Previous NeRF-based dynamic scene representation methods [2, 4–6, 17–22] handle dynamic scenes by learning a mapping from spatiotemporal coordinates to color and density. Unfortunately, these NeRF-based models are constrained in their applications due to low rendering speeds. Recently, 3D Gaussians Splatting [16] has emerged as a novel explicit representation, with many studies [7, 23–27] attempting to model the dynamic scenes based on it. 4D Gaussian Splatting(4DGS) [1] is one of the representatives. It utilizes a set of 4D Gaussian primitives. However, 4DGS often requires a huge redundant number of Gaussians for dynamic scenes. These Gaussians lead to tremendous storage and suboptimal rendering speed. To this end, we focus on analyzing the temporal redundancy of 4DGS [1] in hopes of developing a novel framework to achieve lower storage requirements and higher rendering speeds.

### 2.3 Gaussian Splatting Compression

3D Gaussian-based large-scale scene reconstruction typically requires millions of Gaussians, resulting in the requirement of up to several gigabytes of storage. Therefore, subsequent studies have attempted to tackle these issues. Specifically, Compgs [28] and Compact3D [29] employ vector quantization to store Gaussians within codebooks. Concurrently, inspired by model pruning, some studies [30–35] have proposed criterion to prune Gaussians by a specified ratio. However, compared to 3DGS [16], 4DGS [1] introduces an extra temporal dimension to enable dynamic representation. Previous 3DGS-based methods may therefore be unsuitable for 4DGS. Consequently, we first identify a key limitation leading to this problem, referred as *temporal redundancy*. Furthermore, we propose a novel pruning criterion leveraging spatial-temporal variation, and a temporal filter to achieve more efficient storage requirements and higher rendering speed.

## 3 Preliminary of 4D Gaussian Splatting

Our framework builds on 4D Gaussian Splatting (4DGS) [1], which reconstructs dynamic scenes by optimizing a collection of *anisotropic 4D Gaussian primitives*. For each Gaussian, it is characterized by a 4D mean $\mu = (\mu_x, \mu_y, \mu_z, \mu_t) \in \mathbb{R}^4$ coupled with a covariance matrix $\Sigma \in \mathbb{R}^{4 \times 4}$.

By treating time and space dimensions equally, the 4D covariance matrix $\Sigma$ can be decomposed into a scaling matrix $S_{4D} = (s_x, s_y, s_z, s_t) \in \mathbb{R}^4$ and a rotation matrix $R_{4D} \in \mathbb{R}^{4 \times 4}$. $R_{4D}$ is represented by a pair of left quaternion $q_l \in \mathbb{R}^4$ and right quaternion $q_r \in \mathbb{R}^4$.

During rendering, each 4D Gaussian is decomposed into a conditional 3D Gaussian and a 1D Gaussian at a specific time $t$. Moreover, the conditional 3D Gaussian can be derived from the properties of the multivariate Gaussian with:

$$
\begin{aligned}
\mu_{xyz|t} &= \mu_{1:3} + \Sigma_{1:3,4}\Sigma_{4,4}^{-1}(t - \mu_t) \\
\Sigma_{xyz|t} &= \Sigma_{1:3,1:3} - \Sigma_{1:3,4}\Sigma_{4,4}^{-1}\Sigma_{4,1:3}
\end{aligned}
\tag{1}
$$

Here, $\mu_{1:3} \in \mathbb{R}^3$ and $\Sigma_{1:3,1:3} \in \mathbb{R}^{3 \times 3}$ denote the spatial mean and covariance, while $\mu_t$ and $\Sigma_{4,4}$ are scalars representing the temporal components. To perform rasterization, given a pixel under view $\mathcal{I}$

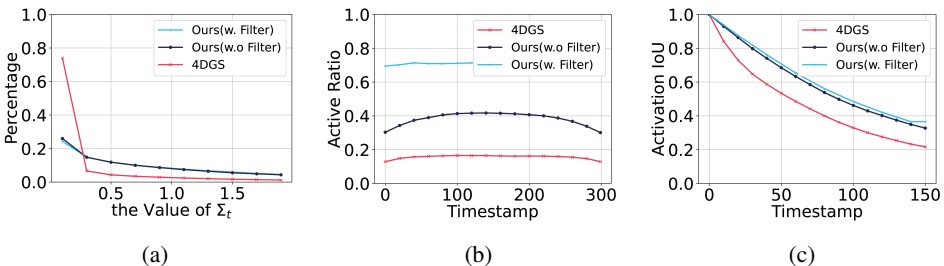

Figure 2: **Temporal redundancy Study.** (a) The $\Sigma_t$ distribution of 4DGS. The red line shows the result of vanilla 4DGS. The other two lines represent our model has effectively reduced the number of transient Gaussians with small $\Sigma_t$. (b) The active ratio during rendering at different timestamps. It demonstrates that most of the computation time is spent on inactive Gaussians in vanilla 4DGS. However, 4DGS-1K can significantly reduce the occurrence of inactive Gaussians during rendering to avoid unnecessary computations. (c) This figure shows the IoU between the set of active Gaussians in the first frame and frame t. It proves that active Gaussians tend to overlap significantly across adjacent frames.

and timestamp $t$, its color $\mathcal{I}(u, v, t)$ can be computed by blending visible Gaussians that are sorted by their depth:

$$\mathcal{I}(u, v, t) = \sum_{i}^{N} c_i(d)\alpha_i \prod_{j=1}^{i-1}(1 - \alpha_j) \tag{2}$$

with

$$\alpha_i = p_i(t)p_t(u, v|t)\sigma_i$$
$$p_i(t) \sim \mathcal{N}(t; \mu_t, \Sigma_{4,4}) \tag{3}$$

where $c_i(d)$ is the color of each Gaussian, and $\alpha_i$ is given by evaluating a 2D Gaussian with covariance $\Sigma_{2D}$ multiplied with a learned per-point opacity $\sigma_i$ and temporal Gaussian distribution $p_i(t)$. In the following discussion, we denote $\Sigma_{4,4}$ as $\Sigma_t$ for simplicity.

**Temporal Redundancy.** Despite achieving high quality, 4DGS requires a huge number of Gaussians to model dynamic scenes. We identify a key limitation leading to this problem: 4DGS represents scenes through temporally independent Gaussians that lack explicit correlation across time. This means that, even static objects are redundantly represented by hundreds of Gaussians, which inconsistently appear or vanish across timesteps. We refer to this phenomenon as *temporal redundancy*. As a result, scenes end up needing more Gaussians than they should, leading to excessive storage demands and suboptimal rendering speeds. In Section 4, we analyze the root causes of this issue and propose a set of solutions to reduce the count of Gaussians.

## 4 Methodology

Our goal is to compress 4DGS by reducing the number of Gaussians while preserving rendering quality. To achieve this, we first analyze the redundancies present in 4DGS, as detailed in Section 4.1. Building on this analysis, we introduce 4DGS-1K in Section 4.2, which incorporates a set of compression techniques designed for 4DGS. 4DGS-1K enables rendering speeds of over 1,000 FPS on modern GPUs.

### 4.1 Understanding Redundancy in 4DGS

This section investigates why 4DGS requires an excessive number of Gaussians to represent dynamic scenes. In particular, we identify two key factors. First, 4DGS models object motion using a large number of transient Gaussians that inconsistently appear and disappear across timesteps, leading to redundant temporal representations. Second, for each frame, only a small fraction of Gaussians actually contribute to the rendering. We discuss those problems below.

**Massive Short-Lifespan Gaussians**. We observe that 4DGS tends to store numerous Gaussians that flicker in time. We refer to these as *Short-Lifespan Gaussians*. To investigate this property,

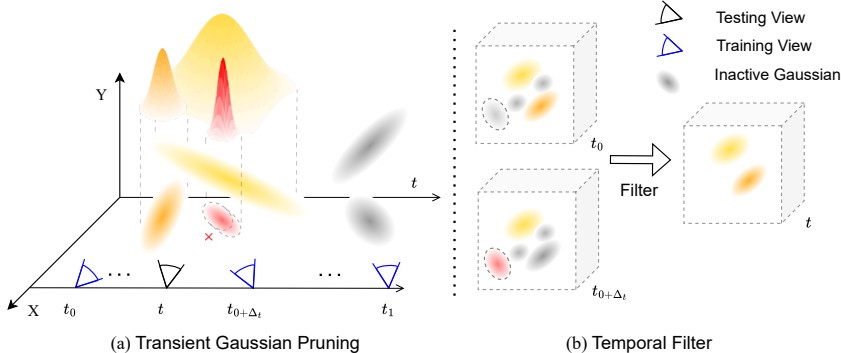

(a) Transient Gaussian Pruning        (b) Temporal Filter

Figure 3: **Overview of 4DGS-1K.** (a) We first calculate the spatial-temporal variation score for each 4D Gaussian on training views, to prune Gaussians with short lifespan (The Red Gaussian). (b) The temporal filter is introduced to filter out inactive Gaussians before the rendering process to alleviate suboptimal rendering speed. At a given timestamp $t$, the set of Gaussians participating in rendering is derived from the two adjacent key-frames, $t_0$ and $t_{0+\Delta_t}$.

we analyze the Gaussians' opacity, which controls visibility. Intuitively, Short-Lifespan Gaussians exhibit an opacity pattern that rapidly increases and then suddenly decreases. In 4DGS, this behavior is typically reflected in the time variance parameter $\Sigma_t$—small $\Sigma_t$ values indicate a short lifespan.

Observations. Specifically, we plot the distribution of $\Sigma_t$ for all Gaussians in the *Sear Steak* scene. As shown in Figure 2a, most of Gaussians has small $\Sigma_t$ values (e.g. 70% have $\Sigma_t < 0.25$).

Therefore, in 4DGS, nearly all Gaussians have a short lifespan. This property leads to high storage needs and slower rendering.

**Inactive Gaussians.** Another finding is that, during the forward rendering, actually, only a small fraction of Gaussians are contributing. Interestingly, active ones tend to overlap significantly across adjacent frames. To quantify this, we introduce two metrics: (1) *Active ratio*. This ratio is defined as the proportion of the total number of active Gaussians across all views at any moment relative to the total number of Gaussians. (2) *Activation Intersection-over-Union (IoU)*. This is computed as IoU between the set of active Gaussians in the first frame and in frame $t$.

Observations. Again, we plot the two metrics from *Sear Steak* scene. As shown in Figure 2b, nearly 85% of Gaussians are inactive at each frame, even though all Gaussians are processed during rendering. Moreover, Figure 2c demonstrates that the active Gaussians remain quite consistent over time, with an IoU above 80% over a 20-frame window.

The inactive gaussians bring a significant issue in 4DGS, because each 4D Gaussian must be decomposed into a 3D Gaussian and a 1D Gaussian before rasterization (see eq. (1)). Therefore, a large portion of computational resources is wasted on inactive Gaussians.

In summary, redundancy in 4DGS comes from massive Short-Lifespan Gaussians and inactive Gaussians. These insights motivate our compression strategy to eliminate redundant computations while preserving rendering quality.

## 4.2 4DGS-1K for Fast Dynamic Scene Rendering

Building on the analysis above, we introduce 4DGS-1K, a suite of compression techniques specifically designed for 4DGS to eliminate redundant Gaussians. As shown in Figure 3, this process involves two key steps. First, we identify and globally prune unimportant Gaussians with low Spatial-Temporal Variation Score in Section 4.2.1. Second, we apply local pruning using a temporal filter to inactive Gaussians that are not needed at each timestep in Section 4.2.2.

### 4.2.1 Pruning with Spatial-Temporal Variation Score

We first prune unimportant 4D Gaussians to improve efficiency. Like 3DGS, we remove those that have a low impact on rendered pixels. Besides, we additionally remove short-lifespan Gaus-

sians—those that persist only briefly over time. To achieve this, we introduce a novel spatial-temporal variation score as the pruning criterion for 4DGS. It is composed of two parts, *spatial score* that measures the Gaussians contributions to the pixels in rendering, and *temporal score* considering the lifespan of Gaussians.

**Spatial score.** Inspired by the previous method [30, 31] and $\alpha$-blending in 3DGS [16], we define the spatial score by aggregating the ray contribution of Gaussian $g_i$ along all rays $r$ across all input images at a given timestamp. It can accurately capture the contribution of each Gaussian to one pixel. Consequently, the spatial contribution score $\mathcal{S}^S$ is obtained by traversing all pixels:

$$\mathcal{S}_i^S = \sum_{k=1}^{NHW} \alpha_i \prod_{j=1}^{i-1}(1 - \alpha_j) \tag{4}$$

where N denotes the number of training views, H, W denote the height and width of the images, and $\alpha_i \prod_{j=1}^{i-1}(1 - \alpha_j)$ reflects the contribution of $i^{th}$ Gaussian to the final color of all pixels according to the alpha composition in eq. (2).

**Temporal score.** It is expected to assign a higher temporal score to Gaussians with a longer lifespan. To quantify this, we compute the *second derivative of* temporal opacity function $p_i(t)$ defined in eq. (3). The second derivative $p_i^{(2)}(t)$ is computed as

$$p_i^{(2)}(t) = (\frac{(t - \mu_t)^2}{\Sigma_t^2} - \frac{1}{\Sigma_t})p_i(t) \tag{5}$$

Intuitively, large second derivative magnitude corresponds to unstable, short-lived Gaussians, while low second derivative indicates smooth, persistent ones.

Moreover, since the second derivative spans the real number domain $\mathbb{R}$, we apply $\tanh$ function to map it to the interval $(0, 1)$. Consequently, the score for opacity variation, $\mathcal{S}_i^{TV}$, of each Gaussian $g_{i,t}$ is expressed as:

$$\mathcal{S}_i^{TV} = \sum_{t=0}^{T} \frac{1}{0.5 \cdot \tanh(\left|p_i^{(2)}(t)\right|) + 0.5}. \tag{6}$$

In addition to the opacity range rate, the volume of 4D Gaussians is necessary to be considered, as described in eq. (1). The volume should be normalized following the method in [30], denoted as $\gamma(\mathcal{S}^{4D}) = Norm(V(\mathcal{S}^{4D}))$. Therefore, the final temporal score $\mathcal{S}_i^T = \mathcal{S}_i^{TV}\gamma(S_i^{4D})$

Finally, by aggregating both spatial and temporal score, the spatial-temporal variation score $\mathcal{S}_i$ can be written as:

$$\mathcal{S}_i = \sum_{t=0}^{T} \mathcal{S}_i^T \mathcal{S}_i^S \tag{7}$$

**Pruning.** All 4D Gaussians are ranked based on their spatial-temporal variation score $\mathcal{S}_i$, and Gaussians with lower scores are pruned to reduce the storage burden of 4DGS [1]. The remaining Gaussians are optimized over a set number of iterations to compensate for minor losses resulting from pruning.

### 4.2.2  Fast rendering with temporal filtering

Our analysis reveals that inactive Gaussians induces unnecessary computations in 4DGS, significantly slowing down rendering. To address this issue, we introduce a temporal filter that dynamically selects active Gaussians. We observed that active Gaussians in adjacent frames overlap considerably (as detailed in Section 4.1), which allows us to share their corresponding masks across a window of frames.

**Key-frame based Temporal Filtering.** Based on this observation, we design a key-frame based temporal filtering for active Gaussians. We select sparse key-frames at even intervals and share their masks with surrounding frames.

Specifically, we select a list of key-frame timestamps $\{t_i\}_{i=0}^{T}$, where $T$ depends on the chosen interval $\Delta_t$. For each $t_i$, we render the images from all training views at current timestamp and calculate the visibility list $\{m_{i,j}\}_{j=1}^{N}$, where $m_{i,j}$ is the visibility mask obtained by eq. (2) from the $j^{th}$ training

Table 1: **Quantitative comparisons on the Neural 3D Video Dataset.**

| Method | PSNR↑ | SSIM↑ | LPIPS↓ | Storage(MB)↓ | FPS↑ | Raster FPS↑ | #Gauss↓ |
|---|---|---|---|---|---|---|---|
| Neural Volume[1][4] | 22.80 | - | 0.295 | - | - | - | - |
| DyNeRF[1][2] | 29.58 | - | 0.083 | 28 | 0.015 | - | - |
| StreamRF[18] | 28.26 | - | - | 5310 | 10.90 | - | - |
| HyperReel[5] | 31.10 | 0.927 | 0.096 | 360 | 2.00 | - | - |
| K-Planes[6] | 31.63 | - | 0.018 | 311 | 0.30 | - | - |
| 4K4D[36] | 21.29 | - | - | 2519 | 290 | - | - |
| Dynamic 3DGS[37] | 30.67 | 0.930 | 0.099 | 2764 | 460 | - | - |
| 4DGaussian[7] | 31.15 | 0.940 | 0.049 | 90 | 30 | - | - |
| E-D3DGS[26] | 31.31 | 0.945 | 0.037 | 35 | 74 | - | - |
| Swift4D[38] | 32.23 | - | 0.043 | 120 | 125 | - | - |
| Grid4D[39] | 31.49 | - | - | 146 | 116 | - | - |
| STG[40] | 32.05 | 0.946 | 0.044 | 200 | 140 | - | - |
| 4D-RotorGS[41] | 31.62 | 0.940 | 0.140 | - | 277 | - | - |
| MEGA[42] | 31.49 | - | 0.056 | 25 | 77 | - | - |
| Compact3D[29] | 31.69 | 0.945 | 0.054 | 15 | 186 | - | - |
| 4DGS[1] | 32.01 | - | 0.055 | - | 114 | - | - |
| 4DGS[2][1] | 31.91 | 0.946 | 0.052 | 2085 | 90 | 118 | 3333160 |
| Ours | 31.88 | 0.946 | 0.052 | 418 | **805** | **1092** | 666632 |
| Ours-PP | 31.87 | 0.944 | 0.053 | **50** | 805 | 1092 | 666632 |

[1] The metrics of the model are tested without "coffee martini" and the resolution is set to $1024 \times 768$.

[2] The retrained model from the official implementation.

Table 2: **Quantitative comparisons on the D-NeRF Dataset.**

| Method | PSNR↑ | SSIM↑ | LPIPS↓ | Storage(MB)↓ | FPS↑ | Raster FPS↑ | #Gauss↓ |
|---|---|---|---|---|---|---|---|
| DNeRF[19] | 29.67 | 0.95 | 0.08 | - | 0.1 | - | - |
| TiNeuVox[43] | 32.67 | 0.97 | 0.04 | - | 1.6 | - | - |
| K-Planes[6] | 31.07 | 0.97 | 0.02 | - | 1.2 | - | - |
| 4DGaussian[7] | 32.99 | 0.97 | 0.05 | 18 | 104 | - | - |
| Deformable3DGS[23] | 40.43 | 0.99 | 0.01 | 27 | 70 | - | 131428 |
| SC-GS[44] | 40.65 | - | - | 28 | 126 | - | - |
| Grid4D[39] | 39.91 | - | - | 93 | 166 | - | - |
| 4D-RotorGS[41] | 34.26 | 0.97 | 0.03 | 112 | 1257 | - | - |
| 4DGS[1] | 34.09 | 0.98 | 0.02 | - | - | - | - |
| 4DGS[1][1] | 32.99 | 0.97 | 0.03 | 278 | 376 | 1232 | 445076 |
| Ours | 33.34 | 0.97 | 0.03 | 42 | **1462** | **2482** | 66460 |
| Ours-PP | 33.37 | 0.97 | 0.03 | **7** | 1462 | 2482 | 66460 |

[1] The retrained model from the official implementation.

viewpoint at timestamp $t_i$ and $N$ is the number of training views at current timestamp. The final set of active Gaussian masks is given by $\left\{ \bigcup_{j=1}^{N} m_{i,j} \right\}_{i=0}^{T}$.

**Filter based Rendering.** To render the images from any viewpoint at a given timestamp $t_{test}$, we consider its two nearest key-frames, denoted as $t_l$ and $t_r$. Then, we perform rasterization while only considering the Gaussians marked by mask $\left\{ \bigcup_{j=1}^{N} m_{i,j} \right\}_{i=l,r}$. This method explicitly filters out inactive Gaussians to speed up rendering.

Note that using long intervals may overlook some Gaussians, reducing rendering quality. Therefore, we fine-tune Gaussians recorded by the masks to compensate for losses.

## 5 Experiment

### 5.1 Experimental Settings

**Datasets.** We utilize two dynamic scene datasets to demonstrate the effectiveness of our method: (1) **Neural 3D Video Dataset (N3V)** [2]. This dataset consists of six dynamic scenes, and the resolution is $2704 \times 2028$. For a fair comparison, we align with previous work [1, 40] by conducting evaluations at a half-resolution of 300 frames. (2) **D-NeRF Dataset** [19]. This dataset is a monocular video dataset comprising eight videos of synthetic scenes. We choose standard test views that originate from novel camera positions not encountered during the training process.

**Evaluation Metrics.** To evaluate the quality of rendering dynamic scenes, we employ several commonly used image quality assessment metrics: Peak Signal-to-Noise Ratio (PSNR), Structural

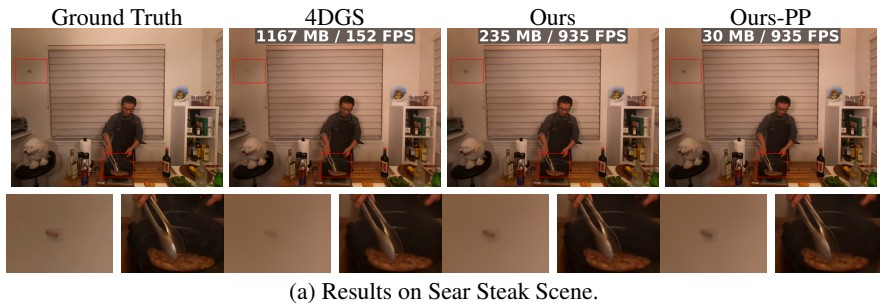

(a) Results on Sear Steak Scene.

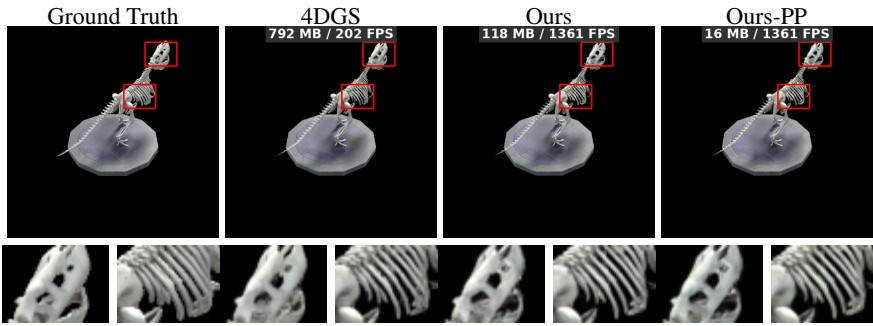

(b) Results on Trex Scene.

Figure 4: **Qualitative comparisons of 4DGS and our method.**

Table 3: **Ablation study of per-component contribution.**

| ID | Method\Dataset | | | PSNR↑ | SSIM↑ | LPIPS↓ | Storage(MB)↓ | FPS↑ | Raster FPS↑ | #Gauss↓ |
|----|---------|---------|-----|-------|-------|--------|--------------|------|-------------|---------|
| | Filter | Pruning | PP | | | | | | | |
| a | vanilla 4DGS[1] | | | 31.91 | 0.9458 | 0.0518 | 2085 | 90 | 118 | 3333160 |
| b | ✓[1,2] | | | 31.51 | 0.9446 | 0.0539 | 2091 | 242 | 561 | 3333160 |
| c | ✓[2] | | | 29.56 | 0.9354 | 0.0605 | 2091 | 300 | 561 | 3333160 |
| d | | ✓ | | 31.92 | 0.9462 | 0.0513 | 417 | 312 | 600 | 666632 |
| e | ✓ | ✓ | | 31.88 | 0.9457 | 0.0524 | 418 | 805 | 1092 | 666632 |
| f | ✓[2] | ✓ | | 31.63 | 0.9452 | 0.0524 | 418 | 789 | 1080 | 666632 |
| g | ✓ | ✓ | ✓ | 31.87 | 0.9444 | 0.0532 | 50 | 805 | 1092 | 666632 |

[1] The result with environment map. [2] The result without finetuning.

Similarity Index Measure (SSIM), and Learned Perceptual Image Patch Similarity (LPIPS) [45]. Following the previous work, LPIPS [45] is computed using AlexNet [46] and VGGNet [47] on the N3V dataset and the D-NeRF dataset, respectively. Moreover, we report the number of Gaussians and storage. To demonstrate the improvement in rendering speed, we report two types of FPS: (1) **FPS.** It considers the entire rendering function. Due to interference from other operations, it can't effectively demonstrate the acceleration achieved by our method. (2) **Raster FPS.** It only considers the rasterization, the most computationally intensive component during rendering.

**Baselines.** Our primary baseline for comparison is 4DGS [1], which serves as the foundation of our model. Moreover, we compare 4DGS-1K with two concurrent works on 4D compression, MEGA [42] and Compact3D [29]. Certainly, we conduct comparisons with 4D-RotorGS [41] which is another form of representation for 4D Gaussian Splatting with the capability for real-time rendering speed and high-fidelity rendering results. In addition, we also compare our work against NeRF-based methods, like Neural Volume [4], DyNeRF [2], StreamRF [18], HyperReel [5], DNeRF [19], K-Planes [6] and 4K4D [36]. Furthermore, other recent competitive Gaussian-based methods are also considered in our comparison, including Dynamic 3DGS [37], STG [40], 4DGaussian [7], E-D3DGS [26], Swift4D [38], Grid4D [39] and SC-GS [44].

**Implementation Details.** Our method is tested in a single RTX 3090 GPU. We train our model following the experiment setting in 4DGS [1]. After training, we perform the pruning and filtering strategy. Then, we fine-tune 4DGS-1K for 5,000 iterations while disabling additional clone/split operations. For pruning strategy, the pruning ratio is set to 80% on the N3V Dataset, and 85% on the D-NeRF Dataset. For the temporal filtering, we set the interval $\Delta_t$ between key-frames to 20 frames

on the N3V Dataset. Considering the varying capture speeds on the D-NeRF dataset, we select $6$ key-frames rather than a specific frame interval. Additionally, to further compress the storage of 4DGS [1], we implement post-processing techniques in our model, denoted as Ours-PP. It includes vector quantization [28] on SH of Gaussians and compressing the mask of filter into bits.

Note that we don't apply environment maps implemented by 4DGS on Coffee Martini and Flame Salmon scenes, which significantly affects the rendering speed. Subsequent results indicate that removing it for 4DGS-1K does not significantly degrade the rendering quality.

## 5.2 Results and Comparisons

**Comparisons on real-world dataset.** Table 1 presents a quantitative evaluation on the N3V dataset. 4DGS-1K achieves rendering quality comparable to the current baseline. Compared to 4DGS [1], we achieve a $41\times$ compression and $9\times$ faster in rendering speed at the cost of a $0.04dB$ reduction in PSNR. In addition, compared to MEGA [42] and Compact3D [29], two concurrent works on 4D compression, the rendering speeds are $10\times$ and $4\times$ faster respectively while maintaining a comparable storage requirement and high quality reconstruction. Moreover, the FPS of 4DGS-1K far exceeds the current state-of-the-art levels. It is nearly twice as fast as the current fastest model, Dynamic 3DGS [37] while requiring only **1%** of the storage size. Additionally, 4DGS-1K achieves better visual quality than that of Dynamic 3DGS [37], with an increase of about $1.2dB$ in PSNR. Compared to the storage-efficient model, E-D3DGS [26] and DyNeRF [2] we achieve an increase of over $0.5dB$ in PSNR and fast rendering speed. Figure 4 offers qualitative comparisons for the Sear Steak, demonstrating that our results contain more vivid details.

**Comparisons on synthetic dataset.** In our experiments, we benchmarked 4DGS-1K against several baselines using the monocular synthetic dataset introduced by D-NeRF [19]. The result is shown in Table 2. Compared to 4DGS [1], our method achieves up to $40\times$ compression and $4\times$ faster rendering speed. It is worth noting that the rendering quality of our model even surpasses that of the original 4DGS, with an increase of about $0.38dB$ in PSNR. Furthermore, our approach exhibits higher rendering quality and smaller storage overhead compared to most Gaussian-based methods. We provide qualitative results in Figure 4 for a more visual assessment.

## 5.3 Ablation Study

To evaluate the contribution of each component, we conducted ablation experiments on the N3V dataset [2]. More ablations are provided in the supplement(See appendix B).

**Pruning.** As shown in Table 3, our pruning strategy achieves $5\times$ compression ratio and $5\times$ faster rasterization speed while slightly improving rendering quality. As shown in Figure 2a, our pruning strategy also reduces the presence of Gaussians with short lifespan. As such, 4DGS-1k processes far fewer unnecessary Gaussians (See Figure 2b) during rendering. Moreover, as shown in Figure 2c, the pruning process expands the range of adjacent frames. It allows larger intervals for the temporal filter.

**Temporal Filtering.** As illustrated in Table 3, the results of b and c are obtained by directly applying the filter without fine-tuning. It proves that this component can enhance the rendering speed of 4DGS. However, as mentioned in Section 4.1, the 4DGS contains a huge number of short lifespan Gaussians. It results in some Gaussians being overlooked in the filter, causing a slight decrease in rendering quality. However, through pruning, most Gaussians are ensured to have long lifespan, making them visible even at large intervals. Therefore, it alleviates the issue of Gaussians being overlooked (See f). Furthermore, appropriate fine-tuning allows the Gaussians in the active Gaussians list to relearn the scene features to compensate for the loss incurred by the temporal filter (See e and f).

## 6 Conclusion

In this paper, we present **4DGS-1K**, a compact and memory-efficient dynamic scene representation capable of running at over 1000 FPS on modern GPUs. We introduce a novel pruning criterion called the spatial-temporal variation score, which eliminates a significant number of redundant Gaussian points in 4DGS, drastically reducing storage requirements. Additionally, we propose a temporal filter that selectively activates only a subset of Gaussians during each frame's rendering. This approach enables our rendering speed to far surpass that of existing baselines. Compared to vanilla 4DGS,

4DGS-1K achieves a $41\times$ reduction in storage and $9\times$ faster rasterization speed while maintaining high-quality reconstruction.

## Acknowledgement

This project is supported by the National Research Foundation, Singapore, under its Medium Sized Center for Advanced Robotics Technology Innovation.

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
