# OpenReview forum: "1000+ FPS 4D Gaussian Splatting for Dynamic Scene Rendering"
_NeurIPS.cc/2025/Conference — NeurIPS 2025 poster_

### Official Review · Reviewer_CKCJ · 2025-06-13

**Clarity:** 3
**Significance:** 3
**Originality:** 3
**Rating:** 4
**Confidence:** 2

**Summary:**

The paper introduces 4DGS-1K, a method to address the inefficiencies of 4D Gaussian Splatting (4DGS) in dynamic scene rendering. The authors identify two key redundancies in 4DGS: (1) Short-Lifespan Gaussians, which are transient and require excessive storage, and (2) Inactive Gaussians, which are processed during rendering but contribute little to the output.

**Questions:**

1. The 1000 FPS claim is striking but requires scrutiny: The "raster FPS" (1092) measures only the rasterization step, while the end-to-end FPS (805) is lower. This discrepancy should be clarified. Is the 1000 FPS achievable in practice, or is it a theoretical peak?
2. The experiments use an RTX 3090, a high-end GPU. Performance on mid-range hardware or edge devices is untested, which could limit real-world applicability.
3. The Spatial-Temporal Variation Score (Eq. 7) combines spatial and temporal terms heuristically. While intuitive, the paper lacks theoretical justification for the aggregation method (e.g., why multiplicative instead of additive?).
4. The pruning threshold (80–85%) is empirical. A sensitivity analysis (e.g., how performance degrades with stricter pruning) would strengthen the method’s robustness.

**Ethical Concerns:**

["NO or VERY MINOR ethics concerns only"]

**Final Justification:**

Thank you for addressing my concerns. While I have no additional issues to raise, I agree with the other reviewer that the generalizability requires stronger justification. I therefore maintain my score.

**Limitations:**

Yes

**Quality:**

2

**Strengths And Weaknesses:**

Strengths：
1. The Spatial-Temporal Variation Score is a novel criterion for pruning Gaussians, combining spatial contribution and temporal stability.
2. The Temporal Filter leverages the observation that active Gaussians overlap across adjacent frames, enabling efficient masking.

Weakness：
The paper introduces meaningful innovations and demonstrates empirical results, but the overstated FPS claim and minor theoretical gaps warrant revisions.

---

> ### Author Rebuttal · Authors · 2025-07-31
>
> We sincerely appreciate the thorough review and insightful feedback provided by each reviewer. The reviewers asked perceptive questions and comments, which are answered in detail in individual responses and have improved our submission.
> > **`>>> Q1:`**
> > **The discrepancy between FPS and raster FPS should be clarified.**
> ---
> **`>>> A1:`** We truly appreciate the reviewer for this excellent question, as it, in fact, touches upon a key aspect of our results.
>
> The two metrics, "raster FPS" and "end-to-end FPS", are presented to **serve complementary purposes**.
>
> - The "raster FPS" (1092 FPS) is reported because it *most directly measures our contribution*: pruning and filtering the number of Gaussians. By isolating the rasterization step, this metric clearly demonstrates the efficiency of our method.
> - The "end-to-end FPS" (805 FPS) represents the *practical rendering speed*. It provides a holistic view by including all stages of the rendering module.
> Both are important, as we aim to demonstrate that we improve the rasterization speed, and consequently, the end-to-end (E2E) speed is also enhanced.
>
> > **`>>> Q2:`**
> > **Performance on mid-range hardware or edge devices.**
> ---
>
> **`>>> A2:`** Thanks for your interest. Our method maintains high rendering speed even on devices with limited computational resources. We have discussed this issue in the Appendix (Section A.2, Lines 778–780). We implement our model on the NVIDIA TITAN X GPU(released in 2015). It still achieves over **200 FPS** on the N3V dataset, significantly outperforming the vanilla 4DGS (20 FPS). It is worth noting that this rendering speed is nearly on par with the performance of other baselines running on an RTX 3090.
>
> > **`>>> Q3:`**
> > **A theoretical justification for the aggregation method (e.g., why multiplicative instead of additive?).**
> ---
> **`>>> A3:`** Thanks for your insightful suggestions. Each 4D Gaussian can be decomposed into a conditional 3D spatial Gaussian and a marginal 1D temporal Gaussian, as demonstrated in 4DGS[1], where $p(x, y, z, t) = p(x, y, z|t)p(t)$. This perspective inspires us to model the spatial and temporal terms as a product. Furthermore, we also report the 'additive' results on the N3V dataset following the default experiment setting. As shown in this table, the product formulation yields higher PSNR (31.92 vs. 31.55) and SSIM (0.9462 vs. 0.9416), and achieves a lower LPIPS score (0.0513 vs. 0.0611). This experiment will be appended to the main paper to support our design of the score.
>
> |   Method   | PSNR$\uparrow$ | SSIM$\uparrow$ | LPIPS$\downarrow$ |
> |:----------:|:--------------:|:--------------:|:------------------:|
> |   Add      |     31.55      |     0.9416     |      0.0611        |
> | Product    |  **31.92**     |  **0.9462**    |   **0.0513**       |
>
>
> [1]Real-time Photorealistic Dynamic Scene Representation and Rendering with 4D Gaussian Splatting
>
> > **`>>> Q4:`**
> > **A sensitivity analysis of pruning threshold.**
> ---
> **`>>> A4:`** The sensitivity analysis is provided in Fig.9 in the Appendices. We analyze the impact of varying pruning ratios on rendering quality. It demonstrates that our spatial-temporal variation score-based pruning can even improve scene rendering quality when the pruning ratio is relatively low. Moreover, it can maintain results comparable to the vanilla 4DGS at higher thresholds.

---

> > ### Comment · Reviewer_CKCJ · 2025-08-08
> > **Thanks**
> >
> > Thank you for addressing my concerns. While I have no additional issues to raise, I agree with the other reviewer that the generalizability requires stronger justification in the revised manuscript. I therefore maintain my positive score.

---

### Official Review · Reviewer_thVU · 2025-06-18

**Clarity:** 3
**Significance:** 2
**Originality:** 3
**Rating:** 4
**Confidence:** 5

**Summary:**

This paper proposes a method to compress 4DGS to maintain rendering quality while improving rendering speed. 4DGS has many inactive and short-lifespan Gaussians, resulting in massive waste of computing resources. To address this problem, the authors design a ranking mechanism combining with temporal and spatial scores to evaluate the contribution of each Gaussian. The Gaussian with low score will be pruned. The experiments demonstrate the effectiveness of proposed methods.

**Questions:**

+ I think SOTA methods, SC-GS and Grid4D, should be included as a baseline in the experiments on the D-NeRF dataset.
+ Lack of training speed evaluation. Will the pruning and filtering processes take a lot of time?
+ A tiny question about the temporal score design. Why not directly use the $|\Sigma_t|$ rather than the second derivative for the temporal score? If possible, could the authors provide some theoretical explanation?

**Ethical Concerns:**

["NO or VERY MINOR ethics concerns only"]

**Final Justification:**

My main concern is the generalizability of the proposed methods which are carefully designed for 4DGS family models. According to the addtional experiments conducted by the authors, the proposed methods are very effective on this models. Although the generalizability might still be an unignorable limitation, the proposed methods are probably useful for the development of 4DGS family models. In general, I decide to raise my rating to boardline accept.

**Limitations:**

Yes, the limitation has been mentioned in the supplementary.

**Paper Formatting Concerns:**

No formatting concerns.

**Quality:**

2

**Strengths And Weaknesses:**

**Strengths**

+ The problem of inactive and short-lifespan Gaussians are clearly analyzed.
+ The proposed methods are very effective in maintaining rendering quality while improving rendering speed.

**Weakness**

+ The proposed methods are carefully designed for 4DGS[1]. This issue severely restricts their application in much higher-performance models, such as Deformable-GS[2], SC-GS[3] and Grid4D[4].
+ When reconstructing scenes with sparse per-frame training views, such as D-NeRF, the proposed methods might lead to obvious rendering quality degradation.

[1] Yang et.al. Real-time Photorealistic Dynamic Scene Representation and Rendering with 4D Gaussian Splatting, ICLR2024.

[2] Yang et.al. Deformable 3D Gaussians for High-Fidelity Monocular Dynamic Scene Reconstruction, CVPR2024.

[3] Huang et.al. SC-GS: Sparse-Controlled Gaussian Splatting for Editable Dynamic Scenes, CVPR2024.

[4] Xu et.al. Grid4D: 4D Decomposed Hash Encoding for High-Fidelity Dynamic Gaussian Splatting, NeurIPS 2024.

---

> ### Author Rebuttal · Authors · 2025-07-31
>
> Thank you for your positive feedback and insightful comments! Below, we respond to address your concern. We welcome further discussion to enhance the clarity and effectiveness of our work.
> > **`>>> W1:`**
> > **Limitations on the application in much higher-performance models, such as Deformable-GS, SC-GS and Grid4D.**
> ---
>
> **`>>> A1:`** We acknowledge that our method was initially designed for 4DGS; however, we argue that our model still demonstrates strong generalizability. While our method is not directly applicable to deformation-based family, it is designed to be **compatible with the entire 4DGS family**. For example, our method is compatible with other works like Temporal Gaussian Hierarchy[1] and FreeTimeGS[2].
>
> [1] Representing Long Volumetric Video with Temporal Gaussian Hierarchy.
>
> [2] FreeTimeGS: Free Gaussian Primitives at Anytime and Anywhere for Dynamic Scene Reconstruction.
>
> >**`>>> W2:`**
> > **Rendering quality degradation on the D-NeRF dataset.**
> ---
>
> **A2:** There is **no degradation** in rendering quality on the D-NeRF datasets. While the results from the original 4DGS [1] are higher than ours, those were obtained under different devices and settings. For a fair comparison, we retrained the official 4DGS code on the D-NeRF dataset using the same protocol. As shown in Table 2, retrained 4DGS achieves 32.99 dB, while our method reaches 33.37 dB. Thus, there is no degradation.
>
> > **`>>> Q1:`**
> > **Baselines such as SC-GS and Grid4D on the D-NeRF dataset.**
> ---
>
> **`>>> A3:`** Thanks for mentioning these excellent works. We compare our method with these methods on the D-NeRF dataset. This table will be appended to Tab.2 in the main paper to provide a more comprehensive comparison. While our method exhibits a reduction in rendering quality compared with these models, it achieves significantly accelerated rendering performance and minimized storage requirements, underscoring its efficiency advantages.
>
> |            Method            | PSNR&nbsp;$\uparrow$ | FPS&nbsp;$\uparrow$ | Storage(MB)&nbsp;$\downarrow$ |
> |:---------------------------:|:--------------------:|:-------------------:|:--------------------------:|
> | SC-GS[1]                     |    **40.65**        |     126                |              28              |
> | Grid4D[3]                     |        39.91     |      166               |                    93        |
> | 4DGS                     |      32.99                |    376                 |          278                 |
> | Ours                     |      33.37                |    **1462**                 |          **7**                 |
>
>
> > **`>>> Q2:`**
> > **Lack of training speed evaluation. Will the pruning and filtering processes take a lot of time?**
> ---
> **`>>> A2:`** No. On the N3V dataset, pruning and filtering processes take nearly 10 minutes. Moreover, 4DGS-1K only takes approximately 30 minutes to fine-tune. This result is provided in Line 773 in the Appendices.
>
> > **`>>> Q3:`**
> > **Why not use the $\left|\Sigma_t\right|$ rather than the second derivative for the temporal score?**
> ---
>
> **`>>> A3:`** We appreciate your valuable insight. Directly using  $\left|\Sigma_t\right|$ for pruning results in a reduction in rendering quality. This has been discussed in the Appendix(See Tab.4, panels (e)). Using $\left|\Sigma_t\right|$ yields a PSNR of 33.47 dB on the *Sear Steak* scene, compared to 33.60 dB for vanilla 4DGS. The PSNR achieved by our method is 33.76 dB. This is because $\left|\Sigma_t\right|$ follows a long-tailed distribution, with its values extending over the entire set of real numbers. As a result, this score disproportionately highlights a minority of large $\left|\Sigma_t\right|$, and diminishes the discriminative capacity of Gaussians with smaller $\left|\Sigma_t\right|$.

---

> > ### Comment · Reviewer_thVU · 2025-08-03
> >
> > Thanks for your answers and clarifications.  I think most of my concerns have been addressed. However, I have the same question as Reviewer nmKh. Could you provide several experimental results to show the generalizability of your methods?

---

> > > ### Author Response · Authors · 2025-08-04
> > >
> > > Thanks for your advice! As suggested, we conducted additional experiments by implementing our method on Spacetime Gaussian (STG) [1], which is another representative work in the 4DGS family.
> > >
> > > **Spacetime Gaussian (STG)**. Specifically, STG uses a temporal radial basis function to model the lifespan of Gaussians. Its temporal opacity can be written as
> > > \begin{equation}
> > > \sigma_i(t)=\sigma^s_i \cdot exp(-s_i^\tau|t-\mu_i^\tau|^2)
> > > \end{equation}
> > > where $i$ denotes the $i$-th Gaussian. $\mu_i^\tau$ is learnable temporal center, $s_i^\tau$ is learnable temporal scaling factor and $\sigma^s_i$ is time-independent spatial opacity.
> > >
> > > This temporal function is directly analogous to the $p
> > > _i(t)$ component in the original 4DGS model, which also leads to a large portion of short-lifespan Gaussians in STG scenes. Accordingly, we compute the Spatial-Temporal Variation Score of each STG in the same way as 4DGS.
> > >
> > > **Experimental Results**. We tested this combined approach (Ours + STG) on the *Cut Roasted Beef* scene from the Neural 3D Video dataset. We set the pruning ratio to 80%, and the interval between key frames was 10 frames. The final results are presented below.
> > > On top of STG, our method reduces storage overhead by $10\times$ and improves rendering speed by $1.78\times$, with a decrease of approximately 0.3 dB in PSNR. Moreover, we also compare our method with a compactness-oriented framework, Compact3D [2], which is **specifically tailored for STG**. By contrast, our method achieves superior rendering quality, improving by approximately $0.18$ dB, and higher rendering speed.
> > >
> > > This experiment confirms that our approach is not limited to a single model but can indeed **be extended to other 4DGS-based methods, achieving similar performance gain**.
> > >
> > > |   Method   | PSNR$\uparrow$ | FPS$\uparrow$ | Storage(MB)$\downarrow$ |
> > > |:----------:|:--------------:|:--------------:|:------------------:|
> > > |   Vanilla STG      |     **33.36**       |   301        |             145  |
> > > |   Compact3D      |     32.89        |     *303*      |             *19*  |
> > > | STG + Ours    |    *33.07*   |   **537**    |     **14**      |
> > >
> > > [1] Spacetime Gaussian Feature Splatting for Real-Time Dynamic View Synthesis.
> > >
> > > [2] Compact 3D Gaussian Splatting for Static and Dynamic Radiance Fields

---

> > > > ### Comment · Reviewer_thVU · 2025-08-05
> > > >
> > > > Thank you very much for your responses. I think my concerns have been addressed, and I have no further questions.

---

> > > > > ### Author Response · Authors · 2025-08-05
> > > > >
> > > > > Dear Reviewer thVU,
> > > > >
> > > > > We sincerely thank you for your thoughtful response and kind support. Your suggestions have been very helpful! If anything remains unclear, please feel free to let us know; we would be happy to clarify further.
> > > > >
> > > > > Thanks again for your time and valuable feedback.
> > > > >
> > > > > Best regards,
> > > > >
> > > > > Authors of Submission 14058

---

### Official Review · Reviewer_nmKh · 2025-06-30

**Clarity:** 2
**Significance:** 2
**Originality:** 3
**Rating:** 4
**Confidence:** 4

**Summary:**

This paper introduces 4DGS-1K, a framework for compressing dynamic 4D Gaussian scene representations. It leverages a spatio-temporal variation score to prune redundant Gaussians and employs keyframe visibility mask sharing to efficiently skip inactive Gaussians during rendering. The proposed method achieves a high compression ratio and fast rendering speeds across both real-world and synthetic datasets.

**Questions:**

1. It remains unclear how inactive Gaussians are actually identified in practice, even considering the brief explanations on lines 156–161 and 217–219. My current understanding is that they are selected via the temporal filtering process, but the details of computing the visibility mask for temporal filtering are not sufficiently explained (Equation 2 appears to be the standard Gaussian rendering equation, with no information about the visibility mask).
2. I would suggest to reorganizing Section 5.3. For example,  element e, and element f mentioned on lines 293–300 have no clear references or connections for the contents in the main paper. If these contain important results or illustrations, the authors should consider moving them to the main paper to improve readability.
3. Do the proposed strategies also improve training speed?

I am willing to increase my score if my concerns and questions are addressed.

**Ethical Concerns:**

["NO or VERY MINOR ethics concerns only"]

**Final Justification:**

The authors have addressed most of my concerns in the rebuttal. I recommend incorporating the additional experiments presented during the discussion period to better demonstrate the method’s generalization capability and revising the manuscript to clarify previously confusing points.

**Limitations:**

Missing social impact discussion.

**Paper Formatting Concerns:**

N.A.

**Quality:**

2

**Strengths And Weaknesses:**

**Strengths**:
1. the major contribution, i.e., the improvement in FPS, of this paper is noticeable compared to the listed baselines.
2. the authors conduct many experiments to illustrate the advantages of the proposed methods.

**Weakness**:
1. It seems that most of the baseline methods used for comparison are drawn from 2024 conferences. I suggest that the authors include comparisons to more recent and competitive works to better demonstrate the advantages of their method. At a minimum (but not limited to), this should include relevant methods such as 4K4D [A], Swift4D [B], and other state-of-the-art video compression techniques.
2. The reported performance improvements appear inconsistent, while the authors do not sufficient explain such discrepancies. In Section C, the authors present gains on selected D-NeRF scenes with reduced floaters; however, the average performance actually drops. Furthermore, while the floaters are attributed to limited training views, it is not clear how the proposed strategies directly address or improve performance specifically in such under-constrained regions.
3. If raster FPS and the number of Gaussians are included as evaluation metrics, could the authors clarify why most of the baselines do not report values for these two metrics?
4. The method is positioned primarily as an incremental improvement over 4DGS, which may limit its generalizability to other dynamic Gaussian frameworks, even though the authors acknowledge this in the limitations. This constraint could reduce the broader significance and impact of the work.

[A] 4K4D: Real-Time 4D View Synthesis at 4K Resolution

[B] Swift4D: Adaptive divide-and-conquer Gaussian Splatting for compact and efficient reconstruction of dynamic scene

---

> ### Author Rebuttal · Authors · 2025-07-31
>
> We would like to thank the reviewer for their insightful feedback and interesting observations. We address the reviewer’s comments below and will include all the feedback in the revised version of the manuscript.
>
> > **`>>> W1:`**
> > **Comparisons with more recent and competitive works such as 4K4D, Swift4D.**
> ---
>
> **`>>> A1:`** Thanks for mentioning these excellent works! As suggested, we compared our method with 4K4D[1], Swift4D[2], Grid4D[3] and DynMF[4] on the N3V dataset. As shown in the table below, the rendering quality achieved by our model exceeds that of the majority of models, except for Swift4D[2]. Furthermore, our model demonstrates substantially superior rendering speed and storage performance. This table will be appended to Tab.1 in the main paper, and will cite these impressive works.
>
> |            Method            | PSNR&nbsp;$\uparrow$ | FPS&nbsp;$\uparrow$ | Storage(MB)&nbsp;$\downarrow$ |
> |:---------------------------:|:--------------------:|:-------------------:|:--------------------------:|
> | 4K4D[1]                         |     21.29       |        `290`             |           2519                 |
> | Swift4D[2]                   |      **32.23**                |    125                 |             `120`              |
> | Grid4D[3]                     |   31.49          |      116               |          146                  |
> | DynMF [4]                     |   31.49          |      197               |          176                  |
> | Ours                     |      `31.87`                |    **805**                 |          **50**                 |
>
> [1] 4K4D: Real-Time 4D View Synthesis at 4K Resolution.
>
> [2] Swift4D: Adaptive divide-and-conquer Gaussian Splatting for compact and efficient reconstruction of dynamic scene.
>
> [3] Grid4D: 4D Decomposed Hash Encoding for High-Fidelity Dynamic Gaussian Splatting.
>
> [4] DynMF: Neural Motion Factorization for Real-time Dynamic View Synthesis with 3D Gaussian Splatting.
>
> > **`>>> W2:`**
> > **Inconsistent performance between Sec.C and average performance. How do the proposed strategies directly improve performance via floaters pruning?**
> ---
>
> **`>>> A2:`**
> * Sorry for any confusion! There is **no inconsistency** between the results. While the results from the original 4DGS [1] are higher than ours, those were obtained under different devices and settings. For a fair comparison, we retrained the official 4DGS code on the D-NeRF dataset using the same protocol. As shown in Table 2, retrained 4DGS achieves 32.99 dB, while our method reaches 33.37 dB. Thus, we claim the improvement.
> * For floaters pruning, floaters are typically characterized by limited spatial extent and short lifespan. Therefore, floaters tend to receive lower scores relative to other Gaussians. They are pruned directly during this stage, which leads to enhanced rendering quality.
>
> [1] Real-time Photorealistic Dynamic Scene Representation and Rendering with 4D Gaussian Splatting
>
> >**`>>> W3:`**
> > **Why do most of the baselines not report values for raster FPS and the number of Gaussians?**
> ---
>
> **`>>> A3:`** We sincerely apologize for any confusion and wish to emphasize that these two evaluation metrics are meaningful within our context. However, they are not applicable when making comparisons with other baselines.
>
> Most baselines represent the motion of Gaussians through deformation fields or trajectory functions. Before rasterization, they need to obtain the Gaussian attributes at the current moment through an MLP or trajectory function $f$. Therefore, for rendering speed, rasterization is not the only factor affecting FPS and cannot be considered in isolation. For the number of Gaussians, we wish to emphasize that each method defines Gaussian attributes differently, resulting in varying storage per Gaussian. Therefore, comparing the number of Gaussians across these baselines is both unfair and not applicable. The general evaluation metric should be the storage footprint of these models.
>
> **Why do we report these two metrics?** These metrics serve primarily as indicators to demonstrate the improvements our method brings to 4DGS. Raster FPS is reported to avoid the influence of other operations within the rendering module, as discussed in Lines 864 to 868. The number of Gaussians is reported to demonstrate the high pruning rate of our method.
>
> Overall, these two metrics are applied and reported solely to 4DGS and 4DGS-1K, and are therefore not reported for the other baselines.
>
> >**`>>> W4:`**
> > **A limited generalizability which could reduce the broader significance and impact of the work.**
> ---
> **`>>> A4:`** Thanks for the question! While our work builds upon 4DGS, we argue it is a significant leap forward, not incremental at all.
>
> In terms of the task, our method introduces **order-of-magnitude improvements** over the existing 4DGS. Our method is the first to achieve over 1000 FPS for dynamic scenes. We deliver an $8×$ rendering speedup and a $41×$ storage reduction over the baseline.
>
> Regarding generalization, dynamic Gaussian methods fall broadly into two families: deformation-based methods and direct 4D Gaussians.
> While our method is not directly applicable to deformation-based family, it is designed to be **compatible with the entire 4DGS family**. For example, our method is compatible with other works like Temporal Gaussian Hierarchy[1] and FreeTimeGS[2].
>
> [1] Representing Long Volumetric Video with Temporal Gaussian Hierarchy.
>
> [2] FreeTimeGS: Free Gaussian Primitives at Anytime and Anywhere for Dynamic Scene Reconstruction.
>
> > **`>>> Q1:`**
> > **How are inactive Gaussians and the visibility mask actually identified?**
> ---
>
> **`>>> A1:`** The visibility mask $m_{i,j}$ from $j^{th}$ training viewpoint at a given timestamp $t_i$ is naturally obtained through the **rasterization**, which has already been implemented in 3DGS[1]. Furthermore, we define inactive Gaussians as the **complement** of the visibility mask, denoted as $\neg~ m_{i,j}$. We apologize for any confusion and will include the relevant description in the main paper.
>
> [1] 3D Gaussian Splatting for Real-Time Radiance Field Rendering.
>
> > **`>>> Q2:`**
> > **Reorganize Section 5.3 to improve readability.**
> ---
> **`>>> A2:`** We appreciate your suggestion. Sec.5.3 discusses the effects of pruning, filtering, and fine-tuning, which may appear somewhat disjointed. We plan to reorganize this section and revise the table layout to improve clarity and readability. Additionally, we would like to clarify that the results in elements (e) and (f) are referred to in Line 300 to demonstrate the effect of fine-tuning on filtering.
>
> > **`>>> Q3:`**
> > **Do the proposed strategies also improve training speed?**
> ---
> **`>>> A3:`** No. Our method does not demonstrate a significant improvement in training speed.

---

> > ### Comment · Reviewer_nmKh · 2025-08-02
> > **Reply to author of 14058**
> >
> > Dear authors,
> >
> > Thank you for your further clarification. I am wondering whether additional experimental results could be provided in combination with state-of-the-art 4DGS-family pipelines. I also noticed that other reviewers (Reviewer B6dT, thVU) have expressed similar concerns regarding the extendability of the proposed strategies. I would suggest that the authors strengthen their discussion on the generalizability of the method by including basic experimental support. For example, would similar performance gains be consistently observed across these pipelines, or might the advantages diminish in such settings? Based on the current rebuttal, this remains unclear to me.
> >
> > Best,
> > Reviewer nmKh

---

> > > ### Author Response · Authors · 2025-08-04
> > >
> > > Thanks for your advice! As suggested, we conducted additional experiments by implementing our method on Spacetime Gaussian (STG) [1], which is another representative work in the 4DGS family.
> > >
> > > **Spacetime Gaussian (STG)**. Specifically, STG uses a temporal radial basis function to model the lifespan of Gaussians. Its temporal opacity can be written as
> > > \begin{equation}
> > > \sigma_i(t)=\sigma^s_i \cdot exp(-s_i^\tau|t-\mu_i^\tau|^2)
> > > \end{equation}
> > > where $i$ denotes the $i$-th Gaussian. $\mu_i^\tau$ is learnable temporal center, $s_i^\tau$ is learnable temporal scaling factor and $\sigma^s_i$ is time-independent spatial opacity.
> > >
> > > This temporal function is directly analogous to the $p
> > > _i(t)$ component in the original 4DGS model, which also leads to a large portion of short-lifespan Gaussians in STG scenes. Accordingly, we compute the Spatial-Temporal Variation Score of each STG in the same way as 4DGS.
> > >
> > > **Experimental Results**. We tested this combined approach (Ours + STG) on the *Cut Roasted Beef* scene from the Neural 3D Video dataset. We set the pruning ratio to 80%, and the interval between key frames was 10 frames. The final results are presented below.
> > > On top of STG, our method reduces storage overhead by $10\times$ and improves rendering speed by $1.78\times$, with a decrease of approximately 0.3 dB in PSNR. Moreover, we also compare our method with a compactness-oriented framework, Compact3D [2], which is **specifically tailored for STG**. By contrast, our method achieves superior rendering quality, improving by approximately $0.18$ dB, and higher rendering speed.
> > >
> > > This experiment confirms that our approach is not limited to a single model but can indeed **be extended to other 4DGS-based methods, achieving similar performance gain**.
> > >
> > > |   Method   | PSNR$\uparrow$ | FPS$\uparrow$ | Storage(MB)$\downarrow$ |
> > > |:----------:|:--------------:|:--------------:|:------------------:|
> > > |   Vanilla STG      |     **33.36**       |   301        |             145  |
> > > |   Compact3D      |     32.89        |     *303*      |             *19*  |
> > > | STG + Ours    |    *33.07*   |   **537**    |     **14**      |
> > >
> > > [1] Spacetime Gaussian Feature Splatting for Real-Time Dynamic View Synthesis.
> > >
> > > [2] Compact 3D Gaussian Splatting for Static and Dynamic Radiance Fields

---

> > > > ### Comment · Reviewer_nmKh · 2025-08-05
> > > > **Reply to authors**
> > > >
> > > > Thank you for conducting the additional experiments. I have no further questions and will raise my score accordingly.

---

> > > > > ### Author Response · Authors · 2025-08-05
> > > > >
> > > > > Dear Reviewer nmKh,
> > > > >
> > > > > Thank you for acknowledging our work and for raising the score. We sincerely appreciate your constructive and detailed feedback, which has greatly contributed to improving the quality of our paper.
> > > > >
> > > > > Thank you once again for your recognition, which is highly valuable and incredibly encouraging to us!
> > > > >
> > > > > Best regards,
> > > > >
> > > > > Authors of Submission 14058

---

### Official Review · Reviewer_B6dT · 2025-07-02

**Clarity:** 3
**Significance:** 3
**Originality:** 3
**Rating:** 4
**Confidence:** 4

**Summary:**

Drawing insight from short-lifespan Gaussians and inactive Gaussians, this paper proposes 4DGS-1K, a novel post-processing method that prunes 4D Gaussian Splatting representations. It achieves a 41× reduction in storage and a 9× increase in rasterization speed on complex dynamic scenes, while maintaining comparable visual quality.

**Questions:**

Please refer to Cons/Questions.

**Ethical Concerns:**

["NO or VERY MINOR ethics concerns only"]

**Final Justification:**

Thanks for the authors' insightful comments. Most problems have been solved. After carefully reading other reviewers’ comments, I recognize the importance of implementing this method on the 4DGS’s family, which could demonstrate the generalization of the proposed method. Therefore, I would like to maintain my initial positive score and recommend integrating this method on more baselines.

**Limitations:**

The limitations are discussed in the supplementary.

**Quality:**

3

**Strengths And Weaknesses:**

**Pros:**
1. The paper is well-written and provides clear insights into 4D Gaussian Splatting representations. The authors identify that short-lifespan and inactive Gaussians significantly increase storage and rendering costs while having minimal impact on visual quality.
2. 4DGS-1K consists of two key components: a spatio-temporal variation score-based pruning strategy and a temporal filter.
3. Using 4DGS-1K, the method achieves substantial storage reduction (~41×) and accelerates rasterization to over 1000 FPS, all while maintaining high-quality rendering.

**Cons/Questions:**
1. Overall, since this is a post-processing method, it may have some negative impact on rendering quality. Has it been considered whether this method could be applied during training to mitigate quality loss?
2. What does NHW mean in Equation 4?
3. Regarding the temporal score, the reviewer is curious about how transient objects in the scene are handled. Could these dynamic elements be mistakenly filtered out?
4. What does “pp” stand for in Table 3? The reviewer recommends that the authors provide an ablation study on the fine-tuning process.
5. The reviewer recommends that the authors add a discussion about implementing the methods on the recent SOTA 4D Gaussian Representations, e.g., Representing Long Volumetric Video with Temporal Gaussian Hierarchy.

---

> ### Author Rebuttal · Authors · 2025-07-31
>
> We thank the reviewer for the constructive comments. We provide our feedback as follows.
>
> > **`>>> Q1:`**
> > **If this method could be applied during training to mitigate quality loss?**
> ---
> **`>>> A1:`** **Yes.** This is feasible. For example, we can periodically prune and update the visibility mask of Gaussians and continue training. In our experiments, fine-tuning the remaining Gaussians can be considered as a part of this process, and it demonstrates a significant alleviation of the rendering quality loss.
>
> > **`>>> Q2:`**
> > **What does NHW mean in Equation 4?**
> ---
>
> **`>>> A2:`** $N$ denotes the number of training views, while $H$ and $W$ denote the height and width of the images, respectively. By $NHW$, we mean all pixels in the training views. In Equation 4, this indicates that aggregating the contributions of each Gaussian to all pixels across all training views. We appreciate your reminder and will revise this in the main paper.
>
> > **`>>> Q3:`**
> > **How are transient objects in the scene handled? Could these dynamic elements be mistakenly filtered out?**
> ---
>
> **`>>> A3:`** Thanks for your interest.
> * We treat static and dynamic objects, including transient objects, within a unified framework. All Gaussians are evaluated together to estimate their scores and record their visibility masks. It is consistent with the insight on 4DGS for modeling dynamic scenes.
> * In fact, the dynamic elements will **not** be filtered out. We provide a visualization of the pruned Gaussians in the *Sear Steak* Scene in the Appendix(See Sec.A.4 and Fig.7). Our method prunes only redundant parts in vanilla 4DGS, while the remaining Gaussians still accurately render dynamic content such as moving hands and pets. This is because dynamic objects are not represented by a single long-lifespan Gaussian but instead by an aggregation of numerous redundant short‑lifespan Gaussians, as we mentioned in Lines 127 to 134. Our pruning strategy only removes redundant and outlier Gaussians, while preserving legitimate dynamic elements like moving objects.
>
>
> > **`>>> Q4:`**
> > **What does "pp" stand for in Table 3? Provide an ablation study on the fine-tuning process.**
> ---
> **`>>> A4:`**
> * The abbreviation “PP” refers to the Post-Processing techniques mentioned in Lines 259–261.
> * We conducted an ablation study on fine-tuning provided in Tab.3. The fine-tuning process helps mitigate the degradation in rendering quality introduced by pruning and filtering. It achieves an improvement in PSNR from 31.63 dB to 31.88 dB. A comprehensive comparison can be found in Table 3, particularly elements (b) and (c), as well as elements (e) and (f).
>
> > **`>>> Q5:`**
> > **Discussion about implementing the methods on the recent SOTA 4D Gaussian Representations, e.g., Representing Long Volumetric Video with Temporal Gaussian Hierarchy.**
> ---
>
> **`>>> A5:`** Thanks for sharing this impressive work and offering this valuable suggestion. We will discuss two current SOTA 4D Gaussian representations, Temporal Gaussian Hierarchy[1] and FreeTimeGS[2]. Meanwhile, it helps demonstrate the generalizability of our method. It will be appended to our paper.
>
> Temporal Gaussian Hierarchy[1] provides a hierarchical organization of 4DGS[3] to represent Gaussians with varying speeds. Therefore, for the pruning strategy, while considering the $\Sigma_t$ of each Gaussian, the hierarchical depth also needs to be taken into account. For filtering, this hierarchical organization provides additional information for this process to exploit. Different intervals can then be assigned to segments at different levels. For instance, smaller intervals can be assigned to the higher-level segments to capture some short‑lifespan Gaussians. This helps us record Gaussians at a finer granularity, thereby mitigating the loss in rendering quality caused by filtering.
>
> For FreeTimeGS, it models the lifespan of Gaussians by defining a unimodal function with a scaling parameter.
> \begin{equation}
> \sigma(t)=exp(-\frac{1}{2}(\frac{t-\mu_t}{s})^2)
> \end{equation}
> Therefore, for the pruning strategy, the temporal score can still be derived from the second derivative of $\sigma(t)$, whereas the spatial scores follow the original processing procedure. As for the filter, FreeTimeGS is likely to share similar properties that active Gaussians in adjacent frames overlap considerably. Therefore, our filtering strategy can still be applied to this method.
>
> [1] Representing Long Volumetric Video with Temporal Gaussian Hierarchy.
>
> [2] FreeTimeGS: Free Gaussian Primitives at Anytime and Anywhere for Dynamic Scene Reconstruction.
>
> [3] Real-time Photorealistic Dynamic Scene Representation and Rendering with 4D Gaussian Splatting.

---

> > ### Comment · Reviewer_B6dT · 2025-08-05
> >
> > Thanks for the authors' insightful comments. Most problems have been solved. After carefully reading other reviewers’ comments, I recognize the importance of implementing this method on the 4DGS’s family, which could demonstrate the generalization of the proposed method. Therefore, I would like to maintain my initial positive score and recommend integrating this method on more baselines.

---

> > > ### Author Response · Authors · 2025-08-05
> > >
> > > Dear Reviewer B6dT,
> > >
> > > Thank you very much for your comments and support! We will carefully incorporate your suggestions to improve the paper.
> > >
> > > If you have any further questions or additional points you’d like us to address, please don’t hesitate to let us know.
> > >
> > > Best regards,
> > >
> > > Authors of Submission 14058

---

### Note · Authors · 2025-08-14

Dear Program Chair, Senior Area Chair, Area Chair, and Reviewers,

We thank all reviewers for the constructive and positive comments:
* Noticeable improvement on storage and FPS(Reviewer B6dT, nmKh, thVU and CKCJ).
* Clear insights and meaningful innovations(Reviewer B6dT, thVU and CKCJ).
* Novel method and criterion(Reviewer B6dT and CKCJ).

We truly appreciate that **reviewers mentioned we have addressed their concerns and raised questions**. Below, we summarize the main points that the reviewers were most concerned about:

* **Generalizability.** As a response, we provide the results of implementing our method on the Spacetime Gaussian (STG). ** It is another representative work in the 4DGS family. Our method effectively reduces storage overhead and improves rendering speed, with only a slight decrease in rendering quality. Meanwhile, it achieves better results than Compact3D, tailored for STG. In future research, we will further explore this aspect to unlock the full potential of our method.

* **Comparisons with more recent works.** We provide comparison results against the recent and competitive baselines on the N3V dataset and D-NeRF dataset, such as 4K4D, Grid4D, and SC-GS. Compared to these excellent works, **our model demonstrates substantially superior rendering speed and storage performance**, while achieving comparable rendering quality.

Finally, we are deeply grateful that reviewers have recognized our improvement and potential to inspire and advance future exploration in dynamic scene reconstruction.

Best regards,

Authors of Submission 14058

---

### Decision · Program_Chairs · 2025-09-17

**Decision:**

Accept (poster)

**Comment:**

In this work, the authors focus on 4DGS, introducing two major elements helping in the rendering speed enhancement: a pruning criterion that removes short-lifespan Gaussians, and a mask for active Gaussians across consecutive frames. The approach is overall simple and, in the tested benchmark, very effective.

Although some of the reviewers have raised concerns about the generalizability of the proposed approach (which should be indicated more explicitly in the paper), the consensus is positive towards the paper. The authors need to incorporate all the extra elements provided in the discussion with the reviewers into the paper.